# Comparison of Two Different Semiquantitative Urinary Dipstick Tests with Albumin-to-Creatinine Ratio for Screening and Classification of Albuminuria According to KDIGO. A Diagnostic Test Study

**DOI:** 10.3390/diagnostics11010081

**Published:** 2021-01-06

**Authors:** Nikolai C. Hodel, Ali Hamad, Klaus Reither, Irene Mndala Kasella, Salim Abdulla, Andreas Schoetzau, Christoph F. R. Hatz, Michael Mayr

**Affiliations:** 1Medical Outpatient Department, University Hospital Basel, 4031 Basel, Switzerland; epi@nikolaihodel.ch; 2Clinical Research Unit, Swiss Tropical and Public Health Institute, 4051 Basel, Switzerland; klaus.reither@swisstph.ch (K.R.); christoph.hatz@unibas.ch (C.F.R.H.); 3Chronic Disease Clinic, Ifakara Health Institute, Bagamoyo P.O. Box 74, Tanzania; ahamad@ihi.or.tz (A.H.); sabdulla@ihi.or.tz (S.A.); 4Faculty of Science, University of Basel, 4001 Basel, Switzerland; info@eudox.ch; 5Outpatient Clinic, Bagamoyo District Hospital, Bagamoyo P.O. Box 29, Tanzania; irenekasella1963@gmail.com; 6Infectiology/Hospital Hygiene Department, Cantonal Hospital, 9007 St. Gallen, Switzerland

**Keywords:** albuminuria, urinary dipstick test, albumin-to-creatinine ratio, ACR, diagnostic test study, Micral-Test^®^, Coumbur9-Test^®^, diagnostic evaluation, point-of-care diagnostics

## Abstract

Background: Semiquantitative dipstick tests are utilized for albuminuria screening. Methods: In a prospective cross-sectional survey, we analyzed the diagnostic test validity of the semiquantitative colorimetric indicator-dye-based Combur9-Test^®^ and the albumin-specific immunochromatographic assay Micral-Test^®^ for the detection of albuminuria, the distribution of the semiquantitative measurements within the albuminuria stages according to KDIGO, and the utility for albuminuria screening compared with an albumin-to-creatinine ratio (ACR) in a walk-in population. Results: In 970 subjects, albuminuria (≥30 mg/g) was detected in 12.7% (95% CI 85.6–96.3%) with the ACR. Sensitivity was 82.9% (95% CI 75.1–89.1%) and 91.9% (95% CI 88.7–96.9%) and specificity 71.5% (95% CI 68.4–74.6%) and 17.5% (95% CI 15.0–20.2%) for the Combur9-Test^®^ and Micral-Test^®^, respectively. Correct classification to KDIGO albuminuria stages A2/A3 with the Combur9-Test^®^ was 15.4%, 51.4%, and 87.9% at cut-offs of 30, 100, and ≥300 mg/dL, and with the Micral-Test^®^ it was 1.8%, 10.5%, and 53.6% at cut-offs of 2, 5, and 10 mg/dL, respectively. Overall, disagreement to KDIGO albuminuria was seen in 27% and 73% with the Combur9-Test^®^ and Micral-Test^®^, respectively. From the total population, 62.5% and 15.3% were correctly ruled out and 2.2% and 1% were missed as false-negatives by the Combur9-Test^®^ and Micral-Test^®^, respectively. Conclusion: Compared to the Combur9-Test^®^, the utility of the Micral-Test^®^ is limited, because the fraction of correctly ruled out patients is small and a large proportion with a positive Micral-Test^®^ require a subsequent ACR conformation test.

## 1. Introduction

The burden of chronic kidney disease (CKD) is increasingly recognized as a global public health problem, with major negative implications on quality of life, premature death, and enormous costs on healthcare systems [1,2]. It is feared that with the ongoing increase in patients affected by the most common CKD risk factors—diabetes [3], hypertension [4], and obesity [5]—CKD prevalence will continue to rise globally, in both developed and developing countries [1,6,7]. Although only a small fraction of patients with CKD are progressing to end-stage renal disease (ESRD), the vast majority are affected by cardiovascular mortality [8,9]. Because early stages of CKD often remain undiagnosed, strategies are required not only for treatment regimens, but also for early detection [9,10]. Albuminuria is a sensitive surrogate marker for kidney impairment and is independent of a reduced estimated glomerular filtration rate (eGFR) associated with increased risk for chronic kidney disease [7,11,12]. According to the 2012 Kidney Disease, Improving Global Outcomes (KDIGO) guidelines, quantitative albumin-to-creatinine ratio (ACR) is the “first priority assessment test” for the evaluation of albuminuria [13]. However, for population screenings, research, and routine clinical check-ups, point-of-care (POC) urine dipstick devices are often utilized, especially in ambulatory or low–middle income settings, where rapid low-cost diagnostics are essential [9,14,15,16]. In clinical practice and according to guidelines for the classification and diagnosis of albuminuria, a finding of semiquantitatively detected albuminuria requires a subsequent quantitative analysis for confirmation and classification to albuminuria stages [13]. Several studies have explored sensitivity and specificity measures of urine dipstick tests and their predictive values for the detection of albuminuria [10,17,18,19,20,21]. However, relatively little is known about the usefulness of semiquantitative urine dipstick tests for albuminuria diagnosis and classification according to KDIGO guidelines [13].

The aim of this study was to assess the validity of a semiquantitative standard colorimetric indicator-dye-based multi dipstick (Combur9-Test^®^) and an albumin-specific immunochromatographic assay (Micral-Test^®^) compared with the quantitative ACR reference diagnostic test [13]. A further aim was to classify the dipstick results according to KDIGO albuminuria stages, and to assess the utility of the semiquantitative dipstick tests for an albuminuria screening in an outpatient clinic (OPC) walk-in population.

## 2. Materials and Methods

### 2.1. Study Population

This is a prospective cross-sectional study performed in a walk-in population of the outpatient clinic of the Bagamoyo District Hospital (BDH) as part of the RenalOne study as previously described [7]. Briefly, the BDH is located in Bagamoyo township and provides care for a semirural population. Bagamoyo district had approximately 300.000 inhabitants in the 2012 census [22]. The OPC was visited on average by 120 (range 41–164) patients daily. For the current study, one consultation hour of the general outpatient ward was designated to ensure a highly standardized procedure. From the newly registered patients of the OPC, 15 to 20 patients per day were consecutively seen. The call up of the patients from the OPC ward was done through the medical staff, without any involvement of the investigators. Pregnant women, patients younger than 18 years, and patients neither able nor willing to provide informed consent were excluded.

### 2.2. Reporting

We followed the STARD 2015 guidelines for reporting diagnostic accuracy studies, as described by Cohen and Korevaar et al. [23].

### 2.3. Measurements and Procedures

All data were collected in a case report form (CRF), translated from English to Swahili. In all participants, past medical histories and smoking status were assessed through interviews, and body weight and height, and blood pressure (BP) were measured and recorded. Glycated hemoglobin A1c (HbA1c) was measured from capillary blood by using a bedside DCA 2000+ Analyzer (Siemens Healthcare Diagnostics, 8047 Zürich, Switzerland). A random clean urine specimen was collected in all patients. Within 30 min after returning the urine specimen to the study team, two different types of semiquantitative dipstick tests and an ACR test were carried out, in order to detect and to determine the degree of albuminuria.

As index test, two semiquantitative urine dipstick tests were performed and recorded during the time of the ACR measurement: the colorimetric dipstick Combour9-Test^®^ (Bayer Diagnostics, 51368 Leverkusen, Germany) and the immunochromatographic dipstick Micral-Test^®^ (Roche Diagnostics, 6343 Rotkreuz, Switzerland). Both urine dipstick tests were carried out manually according to the manufacturer’s instructions, read, interpreted, and recorded by the onsite investigator and full-time technician (N.H.), and supervised by a senior nephrologist (M.M.) [7]. The semiquantitative total urine protein detection with the Combour9-Test^®^ is based on the ability of the protein to change the dye color of an acid–base indicator, forming the anion of 3′,3″,5′,5″-tetrachlorphenol-3,4,5,6-tetrabromsulfophthalein in presence of protein in the urine, resulting in a gradational change of the indicator dye [24]. Although the Combur9-Test^®^ aims for total protein in the urine, the sensitivity to albumin is higher than to other proteins found in the urine; this is explained by the increased affinity of albumin to protons [24,25,26]. The rational in our study to consider a Combur9-Test^®^ value of ≥15–30 mg/dL (1+) as positive is based on the statement in the KDIGO 2012 guidelines that a reagent strip with (1+) protein positivity can substitute a quantitative albuminuria measurement where it not available [13]. Further, it is pointed out that a reagent strip (1+) protein and ACR ≥ 30 mg/g (≥3 mg/mmol) were associated with subsequent risk for CKD progression, acute kidney injury (AKI) [11], cardiovascular mortality, and all-cause mortality in the general population and in populations at risk [13,27,28]. The semiquantitative urinary dipstick Micral-Test^®^ is a chromatographic immunological procedure based on a gold-labelled monoclonal antibody with high specificity to human albumin. The correct handling is of significant importance for the test accuracy. With the correct amounts of fluid (urine) absorbed, the antibody–albumin conjugate is transported to a reactive detection pad. For the Micral-Test^®^, the detection limit is 20 mg/L (2 mg/dL) [24]; cut-off levels are less clear and estimations for cardiovascular mortality are controversial, however several studies have shown that the test validity measures when compared to quantitative methods were strong at ≥20 mg/L (≥2 mg/dL) [29,30]. Further, at this cut-off, a sensitivity of 90% and a specificity of 96% were shown in a laboratory-based screening against the turbidimetric immunoassay [31]. Two index tests with different detection limits were chosen to analyze the influence of detection limits on models of albuminuria screening algorithms.

The reference test was a POC quantitative ACR measurement in a spot urine sample. According to the KDIGO guidelines, the urine ACR measurement is the “first priority test” for the evaluation of albuminuria for a CKD classification [13]. The ACR is a linear continuous value with a cut-off for a moderately increased albuminuria and a relevant risk for CKD at ACR ≥ 30 mg/g (≥3 mg/mmol) [13]. We performed the albumin/creatinine assay on the DCA 2000+ Analyzer (Siemens Healthcare Diagnostics, 8047 Zürich, Switzerland).) in the ambulatory setting of the OPC laboratory. The test cartridges were stored at 4 °C, and the calibration of the DCA 2000+ analyzer was regularly performed and recorded with the calibration card according to the manufacturer’s guide. The albumin/creatinine assay is based on an immunoassay for the creatinine detection and an alkaline colorimetric assay for the creatinine test [32]. All reagents are contained in the test cartridge, all reaction steps, measurements, and calculations are performed automatically, and results are displayed on the instrument screen [32].

In the “Assessment of Performance” section of the United States (U.S.) Food and Drug Administration (FDA) “510(k) Safety and Effectiveness Summary (K963142)”, it is declared that the DCA 2000+ Analyzer and the albumin/creatinine assay were studied in clinical settings and the results were substantially equivalent to results from methods used in clinical laboratory practice [32]. The linear detection range of the albumin/creatinine assay is an ACR of 1 to 2000 mg/g [32].

The Combur9-Test^®^ was read as follows: negative (neg.), (1+/30 mg/dL), (2+/100 mg/dL), (3+/300 mg/dL), and (4+/≥2000 mg/dL) for total protein. The Micral-Test^®^ was read as (neg.), (20 mg/L (2 mg/dL)), (50 mg/L (5 mg/dL)), and (100 mg/L (10 mg/dL)) for albumin. For the reference test, albuminuria was categorized according to KDIGO stages A1 with ACR (1–29 mg/g) normally to mildly increased, A2 (30–299 mg/g) moderately increased, and A3 (≥300 mg/g) severely increased albuminuria [13,33].

### 2.4. Outcomes/Aims

The primary outcome of this study was the assessment of the diagnostic test validity measures of an indicator-dye-based reagent strip (Combur9-Test^®^) and an albumin-sensitive immunological assay (Micral-Test^®^), for the detection of albuminuria, and the distribution of the semiquantitative urine protein evaluation within the albuminuria stages according to KDIGO guidelines [13]. The secondary outcome was a utility analysis of the semiquantitative dipsticks for albuminuria screening.

### 2.5. Statistical Analyses

Statistical analyses were performed using STATA version 14 (StataCorp., College Station, TX, USA). Discrete variables were expressed as counts (percentage), and comparison between groups was done with Pearson’s chi-square test or Fisher’s exact test. Continuous variables were expressed as mean ± standard deviation (SD) if normally distributed or as median and range if not normally distributed, and *t*-test or Mann–Whitney test was used for comparison between groups.

For the test validity analysis, the sensitivity, specificity, and predictive values were calculated, with numbers extracted from the measures of true positives (Combour9-Test^®^ (≥1+) or Micral-Test^®^ (≥2 mg/dL) and ACR ≥ 30 mg/g), true negatives (Combour9-Test^®^ (<1+) or Micral-Test^®^ (<2 mg/dL) and ACR < 30 mg/g), false positives (Combour9-Test^®^ (≥1+) or Micral-Test^®^ (≥2 mg/dL) and ACR < 30 mg/g), and false negatives (Combour9-Test^®^ (<1+) or Micral-Test^®^ (<2 mg/dL) and ACR ≥ 30 mg/g) cases. For a diagnostic test validity with 95% accuracy, a sample size of *n* = 945 was estimated, in a population with an assumed albuminuria prevalence of 10% [6,14]. For the albuminuria screening algorithms, different cut-off levels for positivity were chosen: for scenario A, Combur9-Test^®^ ≥ 1+ (30 mg/dL) and Micral-Test^®^ ≥ 2 mg/dL, and for scenario B, Combur9-Test^®^ ≥ 2+ (100 mg/dL) and Micral-Test^®^ ≥ 5 mg/dL. For scatterplots, data were logarithmically transformed. *p*-values of <0.05 were considered as statistically significant.

## 3. Results

Overall, 1006 patients were recruited (Figure 1). Five patients aged less than 18 years, 19 pregnant women, and eight patients with a missing ACR test were excluded, leaving 974 patients for the final analysis. In an additional four patients, Combur9-Test^®^ and/or Micral-Test^®^ results were missing, and therefore were not included in the comparison between the semiquantitative dipstick tests and the ACR test, leaving *n* = 970 for the assessment of the test validity, the distribution within albuminuria stages, and the semiquantitative dipstick test utility analysis for albuminuria screening algorithms.

### 3.1. Patient Characteristics

The study population consisted of 301 (30%) males and 673 (70%) females (Table 1). Overall *n* = 124 (12.7%; 95% CI 10.6–14.8%) patients had albuminuria with ACR ≥ 30 mg/g. The median age was 37 years (range 18–91 years), median body mass index (BMI) was 24 kg/m^2^ (range 14–53 kg/m^2^), median HbA1c was 5.4% (range 3.9–14%), median hemoglobin was 12.8 g/dL (range 4.1–22 g/dL), and median systolic and diastolic BP were 124 mmHg (range 70–286 mmHg) and 80 mmHg (range 36–150 mmHg), respectively. Patients with albuminuria were older (45 years (range 18–84 years) versus 36 years (range 18–91 years)), had higher HbA1c (5.6% (range 4.2–14%) versus 5.4% (range 3.9–14%), and more often had diabetes (16% versus 5.5%), higher systolic BP 144 mmHg (range 72–286 mmHg) versus 129 mmHg (70–222 mmHg), and higher diastolic BP 90 mmHg (range 42–150 mmHg) versus 80 mmHg (range 36–140 mmHg) than patients without albuminuria (*p* < 0.001 for all). Patients with albuminuria had more often stage I and stage II hypertension (46% versus 24% and 28% versus 9% (*p* < 0.001)), were more often HIV-positive (11% versus 5.9% (*p* = 0.032)), had more often a history of tuberculosis (11% versus 3.8% (*p* = 0.001)), and had more acute infections (14% versus 7.4% (*p* = 0.013)) than patients without albuminuria.

### 3.2. Sensitivity and Specificity Analysis of Semiquantitative Urine Dipstick Tests Compared with Albumin-to-Creatinine Ratio Test

In Figure 2, the correlation of the semiquantitative dipstick test results with the ACR is summarized. For the Combur9-Test (Figure 2A and embedded table), area (a) shows *n* = 102 true positives, with a positive Combur9-Test^®^ and a positive ACR reference test, (b) shows *n* = 241 false positives, with a positive Combur9-Test^®^ and a negative ACR reference test, (c) shows *n* = 21 false negatives, with a negative Combur9-Test^®^ and a positive ACR reference test, and (d) shows *n* = 606 true negatives, with a negative Combur9-Test^®^ and a negative ACR reference test. For the Micral-Test^®^ (Figure 2B and embedded table), area (a) shows *n* = 113 true positives, with a positive Micral-Test^®^ and a positive ACR reference test, (b) shows *n* = 699 false positives, with a positive Micral-Test^®^ and a negative ACR reference test, (c) shows *n* = 10 false negatives, with a negative Micral-Test^®^ and a positive ACR reference test, and (d) shows *n* = 148 true negatives, with a negative Micral-Test^®^ and a negative ACR reference test.

Summarized in Table 2 are the sensitivity, specificity, the positive predictive value (PPV) and the negative predictive value (NVP) from the Combur9-Test^®^ and the Micral-Test^®^ at different cut-off values compared with the ACR reference test. Overall, albuminuria prevalence was 12.7% (*n* = 123/970) with the ACR reference test at a cut-off of ≥30 mg/g. At a Combur9-Test^®^ cut-off level of ≥30 mg/dL (≥1+), the prevalence of Combur9-Test^®^ positive patients was 35.4% (*n* = 343/970). Of those, 10.5% (*n* = 102) also tested positive with the ACR reference test (≥30 mg/g). At this cut-off, the sensitivity of the Combur9-Test^®^ was 82.9% (95% CI: 75.1–89.1%), the specificity 71.5% (95% CI: 68.4–74.6%), the PPV 29.7% (95% CI: 24.9–34.9%), and the NPV 96.7% (95% CI 94.9–97.9%). At a Micral-Test^®^ cut-off level of ≥20 mg/L (≥2 mg/dL), the prevalence of Micral-Test^®^ test positive patients was 83.7% (*n* = 812/970). Of those, 11.6% (*n* = 113) also tested positive with the ACR reference test at a cut-off of ≥30 mg/g. Compared to an ACR reference cut-off of ≥30 mg/g, the sensitivity of the Micral-Test^®^ was 91.9% (95% CI: 85.6–96.3%), the specificity 17.5% (95% CI: 15.0–20.2%), the PPV 13.9% (95% CI: 11.6–16.5%), and the NPV 93.7% (95% CI: 88.7–96.9%). With increasing cut-off level, Combur9-Test^®^ and Micral-Test^®^ have increasing specificity and decreasing sensitivity (Table 2). This is further explored and illustrated in the receiver operator curves (ROC) and the corresponding area under the curve (AUC) values shown in (Figure 2C,D).

### 3.3. Distribution of The Semiquantitative Urine Dipstick Test Results within the KDIGO Albuminuria Classification

Summarized in Figure 3 is the distribution of the semiquantitative dipstick test results within the KDIGO albuminuria classification scheme based on ACR testing (Figure 3A,B). [13]. The semiquantitative Combur9-Test^®^ dipstick was negative in 64.6% (*n* = 627) of the patients (Figure 3A,C). Of those, 96.6% (*n* = 606) corresponded to ACR category A1 and 3.4% (*n* = 21) to A2/A3. The Combur9-Test^®^ was trace-positive (1+) (30 mg/dL) in 24.7% (*n* = 240) of patients. Of those, 84.6% (*n* = 203) corresponded to ACR category A1 and 15.4% (*n* = 37) to A2/A3. The Combur9-Test^®^ was 2+ (100 mg/dL) in 7.2% (*n* = 70) of patients. Of those, 48.6% (*n* = 34) corresponded to ACR category A1 and 51.4% (*n* = 36) to A2/A3. From the 3.4% (*n* = 33) of patients with Combur9-Test^®^ ≥ 300 mg/dL, 12.1% (*n* = 4) corresponded to ACR category A1 and 87.9% (*n* = 29) to A2/A3. Correct classification to KDIGO albuminuria stages A2/A3 with the Combur9-Test^®^ was 15.4%, 51.4%, and 87.9% at cut-offs of 1+ (30 mg/dL), 2+ (100 mg/dL), and ≥3+ (≥300 mg/dL), respectively. Overall, disagreement to KDIGO albuminuria was seen in 27% (*n* = 262) with the Combur9-Test^®^.

The semiquantitative Micral-Test^®^ identified 16.3% (*n* = 158) of the patients as albuminuria-negative (Figure 3B,D). Of those, 93.7% (*n* = 148) corresponded to ACR category A1 and 6.3% (*n* = 10) to A2/A3. The Micral-Test^®^ dipstick identified 46.2% (*n*= 448) of the patients having albuminuria of 2 mg/dL. Of those, 98.2% (*n* = 440) corresponded to ACR category A1 and 1.8% (*n* = 8) to A2/A3. The Micral-Test^®^ was positive at a cut-off level of 5 mg/dL in 21.5% (*n* = 209) of patients. Of those, 89.5% (*n* = 187) corresponded to ACR category A1 and 10.5% (*n* = 22) to A2/A3. The Micral-Test^®^ was positive at the cut-off level 10 mg/dL in 16% (*n* = 155) of patients. Of these, 46.4% (*n* = 72) corresponded to A1 and 53.6% (*n* = 83) to A2/A3. Correct classification with the Micral-Test^®^ was 1.8%, 10.5%, and 53.6% at cut-offs of 2, 5, and 10 mg/dL respectively. Overall, disagreement to KDIGO albuminuria was seen in 73% (*n* = 709) with the Micral-Test^®^.

### 3.4. Agreement of the Micral-Test^®^ with the Combur9-Test^®^

Summarized in Figure 4A is the agreement of the semiquantitative Combur9-Test^®^ with the ACR reference test and the Micral-Test^®^. From the 627 (100%) patients with a negative Combur9-Test^®^, 96.6% (*n* = 606) had a negative and 3.4% (*n* = 21) a positive ACR and 22.8% (*n* = 143) a negative and 77.2% (*n* = 481) a positive Micral-Test^®^. From the 240 (100%) patients with a 1+ (30 mg/dL) Combur9-Test^®^, 84.6% (*n* = 203) had a negative and 15.4% (*n* = 37) a positive ACR and 5.5% (*n* = 13) a negative and 94.5% (*n* = 227) a positive Micral-Test^®^. From the 70 (100%) patients with 2+ (100 mg/dL) positive Combur9-Test^®^, 48.6% (*n* = 34) had a negative and 51.4% (*n* = 36) a positive ACR and 4.3% (*n* = 3) a negative and 95.7% (*n* = 67) a positive Micral-Test^®^. From the *n* = 33 (100%) patients with a ≥3+ (≥300 mg/dL) positive Combur9-Test^®^, 12.1% (*n* = 4) had a negative and 87.9% (*n* = 29) a positive ACR and 100% (*n* = 33) a positive Micral-Test^®^.

Summarized in Figure 4B is the agreement of the semiquantitative Micral-Test^®^ with the ACR reference test and the Combur9-Test^®^. From the 158 (100%) patients with a negative Micral-Test^®^, 93.7% (*n* = 148) had a negative and 6.3% (*n* = 10) a positive ACR and 91% (*n* = 143) a negative and 9% (*n* = 15) a positive Combur9-Test^®^. From the 448 (100%) patients with a 2 mg/dL positive Micral-Test^®^, 98.2% (*n* = 440) had a negative and 1.8% (*n* = 8) a positive ACR and 87.3% (*n* = 391) a negative and 12.7% (*n* = 57) a positive Combur9-Test^®^. From the 209 (100%) patients with a 5 mg/dL positive Micral-Test^®^, 89.5% (*n* = 187) had a negative and 11.5% (*n* = 22) a positive ACR and 31.6% (*n* = 66) a negative and 68.4% (*n* = 143) a positive Combur9-Test^®^. From the *n* = 155 (100%) patients with a 10 mg/dL positive Micral-Test^®^, 46.4% (*n* = 72) had a negative and 53.6% (*n* = 83) a positive ACR and 11% (*n* = 17) a negative and 89% (*n* = 138) a positive Combur9-Test^®^.

### 3.5. Significance of Possible Models for Albuminuria Screening Algorithms Based on Combur9-Test^®^ or Micral-Test^®^ in Clinical Practice

Figure 5 shows the significance of two possible models for albuminuria screening algorithms based on semiquantitative urine dipsticks for a walk-in population with an albuminuria prevalence of approximately 10% [6,14].

In scenario A, the cut-off for Combur9-Test^®^ positivity was set at (1+) ≥30 mg/dL. A negative test had a high NPV of 96% and included 64.6% (*n* = 627/970) of the screened population (Figure 5A). Without further confirmation of a negative test result, 2.2% (*n* = 21/970) and 17% (*n* = 21/123) cases with a relevant albuminuria of the whole population and of all albuminuria-positives would be missed, respectively. Correctly ruled out would be 62.5% (*n* = 606/970) and 71.5% (*n*= 606/847) of the whole population and all albuminuria-negatives, respectively. Of the 343 patients with a positive dipstick Combur9-Test^®^ and a subsequent ACR conformation test, 10.5% (*n* = 102/970) and 83% (*n* = 102/123) of the whole population and of all albuminuria-positives would be confirmed, and 24.8% (*n* = 241/970) and 70.2% (*n* = 241/343) of the whole population and of all Combur9-Test^®^ positives would be over tested, respectively. In summary, the proposed algorithm would mean that 17% of the true positive results would be missed (i.e., 2.2% of the whole population) and that an unnecessary additional test would have to be carried out in 70.2% of patients with a positive screening dipstick test (i.e., in 24.8% of the whole population).

In scenario A, for Micral-Test^®^ positivity, the cut-off was set at ≥2 mg/dL. A negative test had an NPV of 94% and included 16.3% (*n* = 158/970) of the screened population. Without further confirmation of a negative test result, 1% (*n* = 10/970) and 8% (10/123) of cases with a relevant albuminuria of the whole population and of all albuminuria-positives would be missed, respectively. Correctly ruled-out would be 15.3% (*n* = 148/970) of the population and 17.5% (*n* = 148/847) of all albuminuria-negatives, respectively. Of the 83% (*n* = 812/970) with a positive dipstick Micral-Test^®^ and a subsequent ACR conformation test, 11.6% (113/970) of the population and 92% (*n* = 113/123) of all albuminuria-positives would be confirmed, and 72.1% (*n* = 699/970) of the population and 86% (*n* = 699/812) of all Micral-Test^®^ positives would be over tested, respectively. In summary, the proposed algorithm would mean that 8% of the true positive results would be missed (i.e., 1% of the whole population) and that an unnecessary additional test would have to be carried out in 86% of the patients with a positive screening dipstick test (i.e., in 72.1% of the whole population) (Figure 5A).

In scenario B, the cut-off for Combur9-Test^®^ positivity was set at (≥2+) ≥100 mg/dL. A negative test had an NPV of 85.4% and included 89.4% (*n* = 867/970) of the screened population (Figure 5B). Without further confirmation of a negative test result, 6.0% (*n* = 58/970) and 47% (*n* = 48/123) of cases with a relevant albuminuria of the whole population and of all albuminuria-positives would be missed, respectively. Correctly ruled out would be 83.4% (*n* = 809/970) and 95.5% (*n*= 809/847) of the population and all albuminuria-negatives, respectively. Of the *n* = 103 patients with a positive dipstick Combur9-Test^®^ and a subsequent ACR conformation test, 6.7% (*n* = 65/970) and 53% (*n* = 65/123) of the whole population and of all albuminuria-positives would be confirmed, and 3.9% (*n* = 38/970) and 37% (*n* = 38/103) of the whole population and of all Combur9-Test^®^ positives would be over tested, respectively. In summary, the proposed algorithm would mean that 47% of the true positive results would be missed (i.e., 6% of the whole population) and that an unnecessary additional test would have to be carried out in 37% of patients with a positive screening dipstick test (i.e., in 3.9% of the whole population).

In scenario B, for Micral-Test^®^ positivity, the cut-off was set at ≥5 mg/dL. A negative test had an NPV of 98% and included 62.4% (*n* = 606/970) of the screened population. Without further confirmation of a negative test result, 1.9% (*n* = 18/970) and 15% (18/123) of cases with a relevant albuminuria of the whole population and of all albuminuria-positives would be missed, respectively. Correctly ruled out would be 60.6% (*n* = 588/970) of the population and 69.4% (*n* = 588/847) of all albuminuria-negatives, respectively. Of the 37.5% (*n* = 364/970) with a positive dipstick Micral-Test^®^ and a subsequent ACR conformation test, 10.8% (105/970) of the population and 85% (*n* = 105/123) of all albuminuria-positives would be confirmed, and 26.7% (*n* = 259/970) of the population and 71% (*n* = 259/812) of all Micral-Test^®^ positives would be over tested, respectively. In summary, the proposed algorithm would mean that 15% of the true positive results would be missed (i.e., 1.9% of the whole population) and that an unnecessary additional test would have to be carried out in 71% of the patients with a positive screening dipstick test (i.e., in 26.7% of the whole population) (Figure 5A).

## 4. Discussion

We compared the performance of two different urine dipstick tests in nearly 1000 patients. Numerically, both tests had a comparable test validity and performance for test sensitivity and its NPV, but relevant differences in the specificity and the corresponding PPV. This finding is also reflected in the classification to KDIGO albuminuria stages A1–A3 (Figure 3A,B). The Micral-Test^®^ showed weak allocation to all albuminuria stages when positive and was too sensitive to classify the large fraction of subjects without albuminuria to stage A1 (normal to mildly increased). The Combur9-Test^®^ has its limitations but was accurately classifying patients with strong positive (≥3+) results to the corresponding albuminuria stages A2/A3 (moderately/severely increased), and a majority of those subjects without albuminuria correctly to stage A1. Our data illustrate a dose-dependency between the two urine dipstick tests, which leads to a clear concordance with increasing albumin amounts in urine (Figure 4). This consistency makes it clear that both index tests detect the same individuals when albuminuria is present, whereas the immunochromatographic albumin-specific Micral-Test^®^ already detects albuminuria in concentrations in which the standard colorimetric indicator-dye-based multi dipstick Combur9-Test^®^ is still negative.

In the models for albuminuria screening algorithms, we clearly illustrate that both tests have a high accuracy in correctly ruling out patients with no further need for ACR conformation (Figure 5A,B). However, the difference in absolute numbers was significantly different for the two surveyed index tests. The immunochromatographic albumin-specific Micral-Test^®^ had a specificity of 17.5% (*n* = 148/847 of all ACR negatives) and therefore was correctly ruling out only 15% (*n* = 148/970) of the population. While the colorimetric indicator-dye reagent strip Combur9-Test^®^ with a specificity of 71.5% (*n* = 606/847 of all ACR negatives) was four times more (62% of the population (*n* = 606/970)) efficient in correctly ruling out albuminuria-negatives from the population screened. The consequence of the high proportion already ruled out with the reagent strip Combur9-Test^®^ was resulting in a relevant smaller fraction (35% (*n* = 343/970)) of subsequent ACR testing compared with the Micral-Test^®^, where 83% (*n* = 812/970) of the entire population would require an ACR conformation test.

Although the specificity improved with increasing cut-off level (i.e., ≥5 mg/dL or ≥10 mg/dL), the Micral-Test^®^ did not exceed a PPV of over 54%. Therefore, the chance for a correct detection of a clinically relevant albuminuria according to the KDIGO guidelines with the albumin-specific dipstick test was nearly fifty-fifty, even at its highest test positivity level (≥10 mg/dL) [13]. The vast majority (72.1%; *n* = 699/970) of Micral-Test^®^ positive patients were not confirmed to have albuminuria in a ACR test, and therefore were over tested. Similarly, for the Combur9-Test^®^, more than two-thirds (70.3%, *n* = 241/343) of the patients with a positive dipstick test result were not confirmed to have albuminuria in an ACR conformation test. However, the decisive difference is that the overall number for a required ACR conformation test was significantly smaller with the Combur9-Test^®^ (35%; *n* = 343/970) compared with the Micral-Test^®^ (72.1%; *n* = 699/970). The indicator-dye-based Combur9-Test^®^ strip is of good value to correctly rule out patients with no albuminuria, but the semiquantitative detection device has its limitations, with weak test validity measures for ≥1+ (30 mg/dL) and ≥2+ (100 mg/dL) cut-offs, and therefore these quantifications are of little value for albuminuria classification according to KDIGO [13].

If the cut-off level was increased by one level for each index test, the performance of the Combur9-Test^®^ ≥2+ (100 mg/dL), in regard to correctly ruling out patients, further increased to 83.4%, but at the cost of a relevant number of patients that were missed and judged falsely as negatives (47% of all true positives). If the cut-off level for the Micral-Test^®^ was increased to ≥5 mg/dL, the overall distribution of correctly detected, ruled out, missed, and over tested patients was nearly equal to the Combur9-Test^®^ with its regular cut-off levels at ≥1+ (30 mg/dL) Figure 5C(a,d).

With these results from the colorimetric indicator-dye-based urine test strip, we confirm the findings of large population-based studies from across the globe, and several smaller studies in at-risk cohorts, including some from SSA [17,21,36,37,38,39,40]. However, contrary to the opinion of other authors [41], the utility value of the Micral-Test^®^ to correctly rule out patients is only partially correct, because the fraction of correctly ruled out healthy patients was comparatively small, and a large proportion with a positive Micral-Test^®^ would require a subsequent ACR conformation test according to guideline-orientated clinical practice [13].

In our study, we face some limitations. A change in prevalence may lead to different test validity measures and must be considered also for event (albuminuria) probabilities [42]. There are several reasons why albuminuria prevalence might be overestimated with a single-point measurement of ACR. First, transient albuminuria (due to the day-to-day variability) could not be excluded due to the cross-sectional study design with a missing second urine sample for confirmation, but this is the common practice in prevalence studies and therefore the results should be comparable and to a certain degree generalizable [6,14,43]. The second reason why our prevalence could be overestimated is the study setting within an outpatient clinic, enrolling a clientele which could have a higher estimated albuminuria or CKD prevalence, compared with a community-based setting [14,43,44]. Finally, due to the variability of albuminuria during the day, the random collection of urine samples could skew the prevalence rate [45], but this should not influence the finding of the concordance and agreement between index and reference tests. Both semiquantitative urine dipstick tests are dependent on urine concentration and the urine specific gravity [46], which was not assessed in our laboratory. Further, the albuminuria screening algorithm models were built under the assumption that we detected patients with persistent and not transient albuminuria. However, a strength of our study remains that the index dipstick tests and the ACR reference test were carried out simultaneously from the same urine specimen, supporting the concordance and agreement of the tests.

## 5. Conclusions

The two semiquantitative index tests differ strongly in specificity, especially at their lowest positivity levels. Compared to the Combur9-Test^®^, the more expensive and more specific Micral-Test^®^ does not add any significant benefit to clinical practice and would require a much higher rate of unnecessary subsequent ACR clarification testing, in order to diagnose albuminuria according to guideline recommendations [13]. The Combur9-Test^®^ with the ability to rule out albuminuria accurately, without a relevant loss of missed albuminuria cases, is therefore a valid initial diagnostic tool to screen for albuminuria and CKD in a walk-in population with an assumed albuminuria prevalence of 10%. These findings are of importance considering the distribution of resources in clinical practice and research settings, where upon a positive dipstick test result a subsequent ACR conformation test should be executed.

## Figures and Tables

**Figure 1 diagnostics-11-00081-f001:**
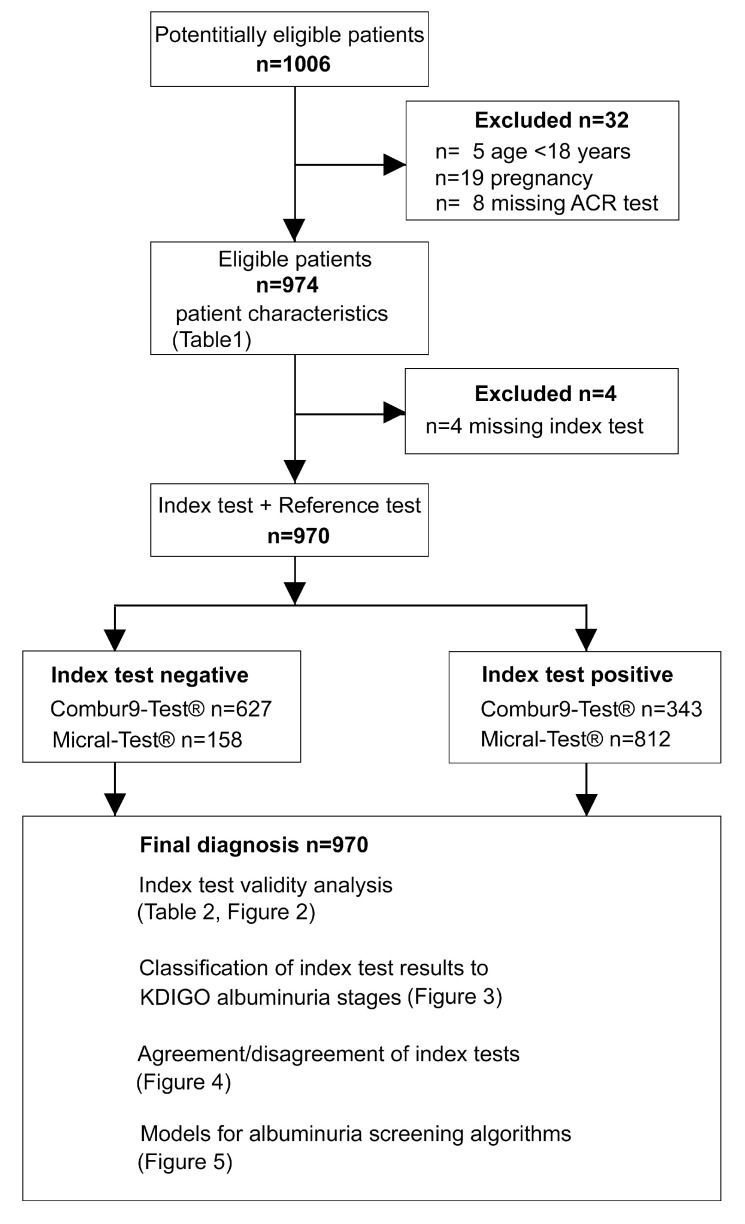
Study flow: ACR: albumin-to creatinine ratio; KDIGO: Kidney Disease Improving Global Outcome.

**Figure 2 diagnostics-11-00081-f002:**
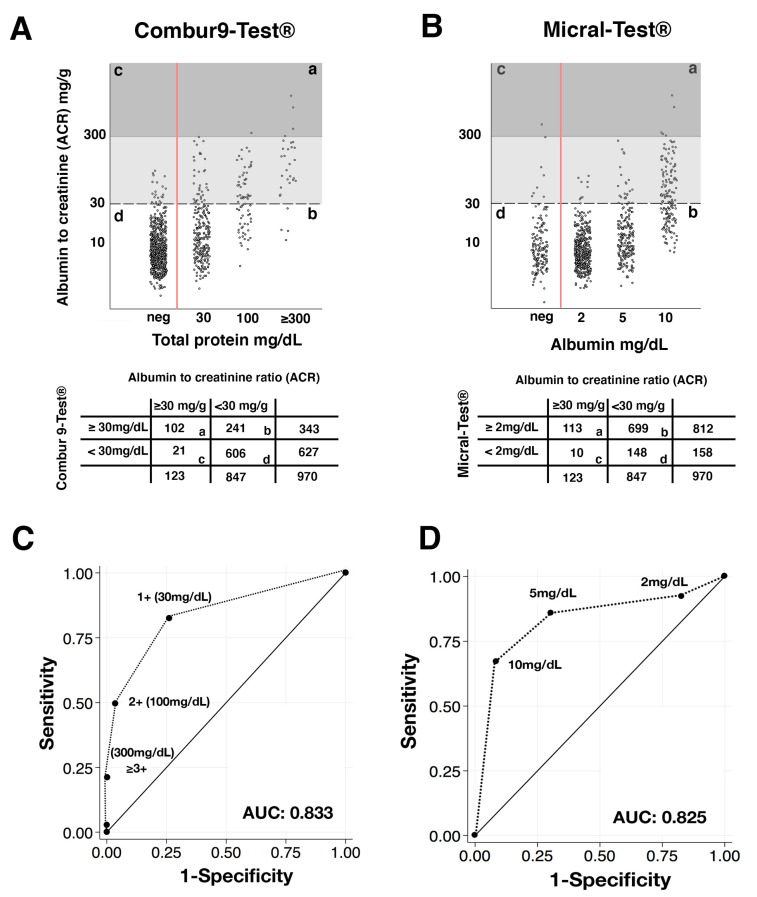
Correlation and agreement of the colorimetric indicator-dye-based urine Combur9-Test^®^ and the immunochromatographic albumin-specific urine Micral-Test^®^ with the albumin-to-creatinine ratio (ACR) reference test and distribution within KDIGO albuminuria classification. (**A**,**B**) with embedded tables: Correlation of Combur9-Test^®^ and Micral-Test^®^ with ACR: (a) dipstick positive and ACR ≥ 30 mg/g; (b) dipstick positive and ACR <30 mg/g; (c) dipstick negative and ACR ≥ 30 mg/g; (d) dipstick negative and ACR < 30 mg/g. Dashed horizontal black line: albuminuria cut-off of ACR for positive testing at 30 mg/g; white area: albuminuria < 30 mg/g; light grey area: albuminuria 30–299 mg/g; dark grey area: albuminuria ≥ 300 mg/g; red line: dipstick negative versus positive; (**C**,**D**): Receiver operator curves (ROC) of index dipstick tests against ACR reference test with cut-off ≥30 mg/g for albuminuria. (**C**) Combur9-Test^®^ versus ACR reference test; (**D**) Micral-Test^®^ versus ACR reference test; AUC: area under the curve; black line: random classifier; dotted line: test classifier.

**Figure 3 diagnostics-11-00081-f003:**
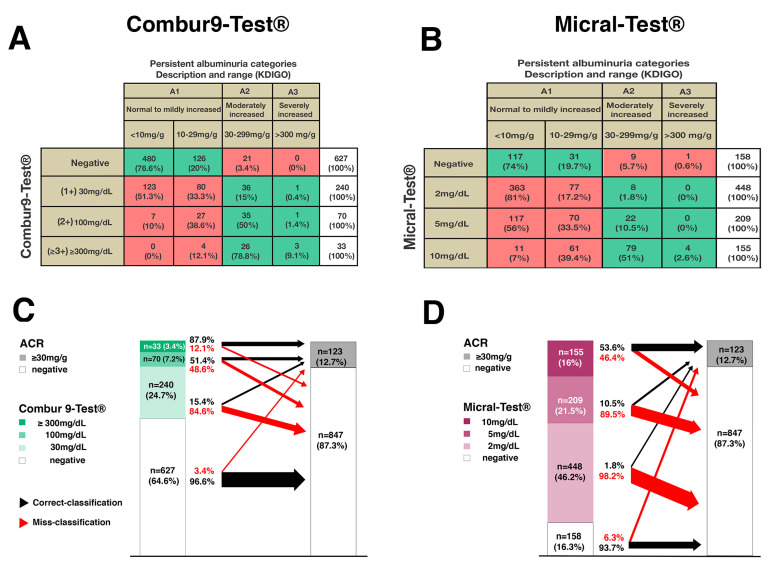
Distribution within KDIGO albuminuria classification: albuminuria stages A1: normally to mildly increased (<10–29 mg/g), A2: moderately increased (30–299 mg/g), A3: severely increased (≥300 mg/g); green areas: true-negative and true-positive classifications; red areas: false-positive and false-negative classifications; (**A**) total disagreement (red areas) and agreement (green areas) between index test (Combur9-Test^®^) and ACR test in *n* = 262 (27%) and *n* = 708 (73%), respectively; (**B**) total disagreement (red areas) and agreement (green areas) between index test (Micral-Test^®^) and ACR test in *n* = 709 (73%) and *n* = 261 (27%), respectively. (**C**,**D**): agreement/disagreement of (**C**) Combur9-Test^®^ and (**D**) Micral-Test^®^ with ACR at a cut-off of 30 mg/g: black arrow: agreement; red arrow: disagreement; arrow widths are proportional to the percentage of agreement/disagreement; KDIGO: Kidney Disease Improving Global Outcome [13]; ACR: albumin-to-creatinine ratio.

**Figure 4 diagnostics-11-00081-f004:**
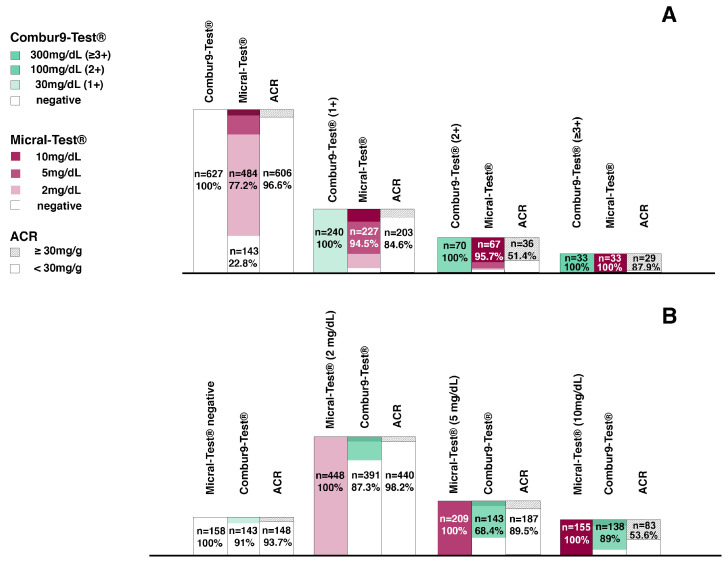
Agreement of Combur9-Test^®^ and Micral-Test^®^ with the albumin-to-creatinine ratio (ACR). ACR as reference test with cut-off ≥30mg/g. (**A**) Colorimetric urinary dipstick Combur9-Test^®^ results compared to the albumin-to-creatinine ratio and to the immunochromatographic Micral-Test^®^; (**B**) immunochromatographic Micral-Test^®^ results compared to the albumin-to-creatinine ratio and to the colorimetric urinary dipstick Combur9-Test^®^; ACR: albumin-to-creatinine ratio.

**Figure 5 diagnostics-11-00081-f005:**
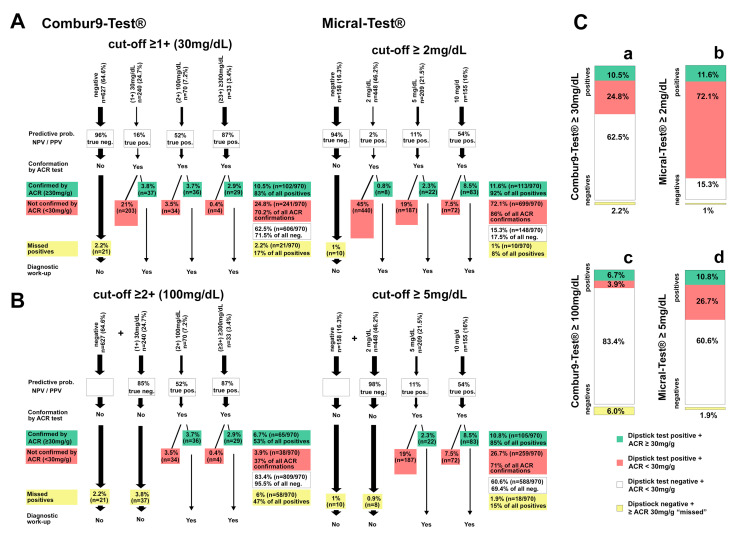
Models for albuminuria screening algorithms with Combur9-Test^®^ and Micral-Test^®^ in 970 patients (reference test: albumin-to-creatinine ratio (ACR) with a cut-off of ≥30 mg/g): (**A**) cut-off for negative index tests: Combur9-Test^®^ cut-off ≥1+ (≥30 mg/dL) and Micral-Test^®^ cut-off ≥2 mg/dL; NPV all negative Combur9-Test^®^ and Micral-Test^®^; (**B**) cut-off for negative index tests: Combur9-Test^®^ cut-off ≥2+ (≥100 mg/dL) and Micral-Test^®^ cut-off ≥5 mg/dL; NPV all patients ≤1+ Combur9-Test^®^, all patients ≤5 mg/dL Micral-Test^®^; true neg: true negatives with index test and confirmation test negative; true pos: true positives with index test and confirmation test positive; PPV: positive predictive value; (**C**) Distribution of correctly confirmed Combur9-Test^®^ and Micral-Test^®^ by an ACR at a cut-off of 30 mg/g; (a) Combur9-Test^®^ cut-off ≥1+ (≥30 mg/dL) (b) Micral-Test^®^ cut-off ≥2 mg/dL, (c) Combur9-Test^®^ cut-off ≥2+ (≥100 mg/dL) (d) Micral-Test^®^ cut-off ≥5 mg/dL for positivity; NPV: negative predictive value; ACR: albumin-to-creatinine ratio.

**Table 1 diagnostics-11-00081-t001:** Patient characteristics.

	*n* (Missing)	Overall	ACR < 30 mg/g	ACR ≥ 30 mg/g	*p*-Value
Overall	974	974	850 (87.3%)	124 (12.7%) (^#^ CI 95%: 10.6–14.8%)	
Male vs.	974	301 (30%)	261 (87%)	40 (13%)	
Female		673 (70%)	589 (88%)	84 (12%)	0.755 *
Age (years)	968 (6)	37 (18–91)	36 (18–91)	45 (18–84)	<0.001 °
BMI (kg/m^2^) ^†^	971(3)	24 (14–53)	24.1 (14–53)	23.8 (15–43)	0.651 °
BP systolic (mmHg) ^‡^	971 (3)	124 (70–286)	129 (70–222)	144 (72–286)	<0.001 °
BP diastolic (mmHg) ^§^ Hypertension stage I ^•^ Hypertension stage II ^••^ HbA1c (%) ^¶^ HbA1c ≥ 6.5% Diabetes ^Φ^ Hemoglobin (g/dL)Anemia (WHO) ** Acute infection ^##^ HIV positive ^ƒ^ History of tuberculosis	971 (3) 971 (3) 971 (3) 965 (9) 965 (9) 974 910 (64) 903 (71) 974 974 972 (2)	80 (36–150) 261 (27%) 115 (12%) 5.4 (3.9–14) 63 (6.5%) 67 (6.8%) 12.8 (4.1–22) 321 (36%) 81 (8.3%) 64 (6.6%) 46 (4.7%)	80 (36–140) 204 (24%) 80 (9%) 5.4 (3.9–14) 44 (42%) 47 (5.5%) 12.8 (4.1–22) 275 (35%) 63 (7.4%) 50 (5.9%) 32 (3.8%)	90 (42–150) 57 (46%) 35 (28%) 5.6 (4.2–14) 19 (15%) 20 (16%) 12.5 (6.3–16) 46 (43%) 18 (14%) 14 (11%) 14 (11%)	<0.001 ° <0.001 * <0.001 * 0.004 ° <0.001 * <0.001 * 0.017 ° 0.084 * 0.013 * 0.032 * 0.001 *
History of Smoking	974	74 (7.6%)	65 (7.7%)	9 (7.3%)	1.000 *

Data are displayed as counts and (percent) or median and (range); ° Mann–Whitney-U (rank sum) test, * Fisher’s exact test; ^#^ CI: confidence interval; ^†^ BMI: body mass index (kg/m^2^); **^‡^** BP systolic: blood pressure systolic, ^§^ BP diastolic: blood pressure diastolic; ^•^ Hypertension stage I: BP systolic ≥ 140 mmHg and/or BP diastolic ≥ 90 mmHg; ^••^ Hypertension stage II: BP systolic ≥ 160 mmHg and/or BP diastolic ≥ 100 mmHg [34]; ^¶^ HbA1c: glycated hemoglobin; ^Φ^ Diabetes: patients with a HbA1c ≥ 6.5% and/or a history of diabetes and/or antidiabetic medication. ** Anemia according to WHO [35]: <12 g/dL in female, <13 g/dL in male; ^#^^#^ Acute infection: acute systemic infection/inflammation or possible urinary tract infection (UTI), defined as body temperature of ≥38.5 °C (armpit), acute malaria, acute tuberculosis, leukocyte count >20/high power field microscopy (HPF) in urinary sediment or newly positive tested HIV cases; ^ƒ^ HIV positive: 44 patients were diagnosed with HIV by testing within the study, 22 patients had a history of HIV and 20 of them were on antiretroviral therapy.

**Table 2 diagnostics-11-00081-t002:** Sensitivity, specificity, and negative and positive predictive values of semiquantitative colorimetric indicator-dye-based Combur9-Test^®^ and immunochromatographic Micral-Test^®^ compared with albumin-to-creatinine ratio (ACR) reference test.

Test	*n* (%)	*n* (%) + ACR ≥30 mg/g	Sensitivity % (CI 95%)	Specificity % (CI 95%)	PPV (CI 95%)	NPV (CI 95%)
ACR ≥30 mg/g	123 (12.7%)					
Combur9-Test^®^	970 (100%)					
≥30 mg/dL	343 (35.4%)	102 (10.5%)	82.9% (75.1–89.1%) *n* = 102/123 * *n* = 21	71.5% (68.4–74.6%) *n* = 606/847 ° *n* = 241	29.7% (24.9–34.9%) *n* = 102/343 ° *n* = 241	96.7% (94.9–97.9%) *n* = 606/627 * *n* = 21
≥100 mg/dL	102 (10.5%)	65 (6.7%)	52.8% (44.1–62.2%) *n* = 65/123 * *n* = 58	95.5% (93.5–96.5%) *n* = 809/847 ° *n* = 38	63.1% (51.8–70.9%) *n* = 65/103 ° *n* = 38	93.3% (91.4–94.9%) *n* = 809/867 * *n* = 58
≥300 mg/dL	33 (3.7%)	29 (3.3%)	26.0% (17.0–32.7%) *n* = 29/123 * *n* = 94	99.5% (98.3–99.7%) *n* = 843/847 ° *n* = 4	87.9% (64.8–92.0%) *n* = 29/33 ° *n* = 4	90.0% (87.9–91.8%) *n* = 843/937 * *n* = 94
Micral–Test^®^	970 (100%)					
≥2 mg/dL	812 (83.7%)	113 (11.6%)	91.9% (85.6–96.3%) *n* = 113/123 * *n* = 10	17.5% (15.0–20.2%) *n* = 148/847 ° *n* = 699	13.9% (11.6–16.5%) *n* = 113/812 ° *n* = 699	93.7% (88.7–96.9%) *n* = 148/158 * *n* = 10
≥5 mg/dL	364 (37.4%)	105 (10.8%)	85.4% (77.9–91.1%) *n* = 105/123 * *n* = 18	69.4% (66.2–72.5%) *n* = 588/847 ° *n* = 259	28.9% (24.3–33.8%) *n* = 105/364 ° *n* = 259	97.0% (95.4–98.2%) *n* = 588/606 * *n* = 18
≥10 mg/dL	155 (15.9%)	83 (8.6%)	67.5% (58.5–75.7%) *n* = 83/123 * *n* = 40	91.5% (89.4–93.3%) *n* = 775/847 ° *n* = 72	53.5% (45.4–61.6%) *n* = 83/155 ° *n* = 72	95.1% (93.4–96.5%) *n* = 775/815 * *n* = 94

ACR: albumin-to-creatinine ratio; CI: confidence interval; PPV: positive predictive value; NPV: negative predictive value; * false negatives; ° false positives; Combur9-Test^®^ total protein urine dipstick in mg/dL; Micral-Test^®^ albumin-specific urine dipstick in mg/dL.

## Data Availability

The data presented in this study are available on request from the corresponding author.

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
