# Peer review of "Comparison of Two Different Semiquantitative Urinary Dipstick Tests with Albumin-to-Creatinine Ratio for Screening and Classification of Albuminuria According to KDIGO. A Diagnostic Test Study"

_diagnostics, 2021, doi:10.3390/diagnostics11010081_

Round 1

Reviewer 1 Report

In this paper, Nikolai C. Hodel et al. compared two urinary dipstick tests, with a gold standard-ACR assay, to assess the clinical utility of proteinuria testing in the general population. This is an important study investigating the sensitivity and specificity of two commercially-available dipstick tests, in a large group of patients (n=974). Another advantage of this paper is a rational laboratory algorithm for albuminuria testing in a relation to the KDIGO guidelines.

While this at first glance appears relevant and clear, upon closer look, some issues have not been addressed.

Major:

  1. While all the statistics were done for the general population only, this is a major limitation of this study. Giving separate analysis for diabetes, hypertension, autoimmunity, or infection groups could be beneficial, as for instance in cardiovascular patients even low albuminuria was significant (Nephron 2018;140:169–174 DOI: 10.1159/000490954). For this reason, it would be interesting to validate dipstick test results according to the patient’s status. So, the patient’s clinical characterization should be given in Table 1.
  2. It is not clear what kind of samples were used for testing, random or first-morning specimen (page 2, line 87-89)? If it’s not standardized in this study, would it affect results? Another concern was the urine dipstick methods selection. Since 30mg/g is considered a clinically relevant cut-off, why did the authors used tests with two different detection limits? What were the selection criteria for this issue? This should be clearly stated in the method section. (Adv Chronic Kidney Dis 2011 Jul;18(4):243-8. doi: 10.1053/j.ackd.2011.03.002)
  3. Because dipstick assays were carried out manually (page 3, line 93-96), basic method validation should be provided. This could be a comparison of negative and (1+) dipstick with ACR results. If applicable, standard addition methodology would be desirable (Journal of AOAC INTERNATIONAL, Volume 78, Issue 2, 1 March 1995, Pages 471–476, https://doi.org/10.1093/jaoac/78.2.471). The validation should prove that the manual readout was close to the manufacturer’s detection limit. Moreover, a single dipstick test could not be sufficient to detect proteinuria, and a double test would be needed to match with ACR result (Int J Environ Res Public Health. 2020 Jun; 17(12): 4195. doi: 10.3390/ijerph17124195).

Minor:

  1. Apart from NPV and PPV values, ROC analysis would enrich the results of your study.
  2. Please consider citing: Nephron 2018;140:169–174 DOI: 10.1159/000490954 ; Adv Chronic Kidney Dis 2011 Jul;18(4):243-8. doi: 10.1053/j.ackd.2011.03.002 ; Int J Environ Res Public Health. 2020 Jun; 17(12): 4195. doi: 10.3390/ijerph17124195

Author Response

Review I

In this paper, Nikolai C. Hodel et al. compared two urinary dipstick tests, with a gold standard-ACR assay, to assess the clinical utility of proteinuria testing in the general population. This is an important study investigating the sensitivity and specificity of two commercially-available dipstick tests, in a large group of patients (n=974). Another advantage of this paper is a rational laboratory algorithm for albuminuria testing in a relation to the KDIGO guidelines.

While this at first glance appears relevant and clear, upon closer look, some issues have not been addressed.

Major:

  1. While all the statistics were done for the general population only, this is a major limitation of this study. Giving separate analysis for diabetes, hypertension, autoimmunity, or infection groups could be beneficial, as for instance in cardiovascular patients even low albuminuria was significant (Nephron 2018;140:169–174 DOI: 10.1159/000490954). For this reason, it would be interesting to validate dipstick test results according to the patient’s status. So, the patient’s clinical characterization should be given in Table 1.

We appreciate the comment and thank the reviewer for the valuable inputs. It is correct that we performed an analysis in a general population of an outpatient clinic. As suggested by the reviewer, we have added additional patient characteristics to table 1 in order to obtain a better overview of the population. We have specifically added cardiovascular risk factors for albuminuria and infectious diseases relevant in the context of sub-Saharan Africa.  Due to the small number of subjects in the sub-populations, we did not perform sub-analyses of these specific patient groups in this study, as the aim of the study was to validate the urine dipstick tests in a population of an outpatient clinic.

To table 1 (patient characteristics) on page 6 line 210-211 we added: Univariate analysis for the outcome albuminuria ACR ≥ 30mg/g: Hypertensive blood pressure stage I ≥ 90/140 mmHg, ••Hypertensive blood pressure stage II ≥ 100/160 mmHg; HbA1c (%); HbA1c ≥6.5%; Patients with diabetesF; Hemoglobine (g/dL) and patients with anemia according to WHO **, acute infection#, HIV ƒ and a history of tuberculosis.

To the caption of table 1 page 6 line 213-219 we added: “Hypertension stage I: BP systolic ≥ 140mmHg and/or  BP diastolic ≥ 90mmHg ••Hypertension stage II: BP systolic ≥ 160mmHg and/or BP diastolic ≥ 100mmHg [1]; HbA1c: Glycated hemoglobin; FDiabetes: patients with a HbA1c ≥ 6.5% and/or a history of diabetes and/or anti-diabetic medication. ** Anemia according to WHO [2]: <12g/dL in female, <13g/dL in male; #Acute infection: acute systemic infection/inflammation or possible UTI, defined as body temperature of ≥38.5°C (armpit), acute malaria, acute tuberculosis, leukocyte count >20/HPF in urinary sediment or newly positive tested HIV cases; ƒHIV positive: 44 patients were diagnosed with HIV by testing within the study, 22 patients had a history of HIV and 20 of them were on antiretroviral therapy”

In the method section on page 2 line 89-91 we added: “HbA1c was measured from capillary blood by using a bed-side DCA 2000+ Analyzer (Siemens Healthcare Diagnostics).”

In the results section paragraph 3.1 on page 5/6 line 196-208 were modified as follows:”The study population consisted of 301 (30%) males and 673 (70%) females (table 1). Overall n=124 (12.7%; 95% CI 10.6-14.8%) patients had albuminuria with ACR ≥30mg/g. Median age was 37 years (range 18-91 years), median BMI 24 kg/m2 (range 14-53 kg/m2), median HbA1c 5.4% (range 3.9-14%), median hemoglobin 12.8 g/dL (range 4.1-22 g/dL), and median systolic and diastolic BP 124 mmHg (range 70-286 mmHg) and 80 mmHg (range 36-150 mmHg), respectively. Patients with albuminuria were older (45 years (range 18-84 years) versus 36 years (range 18-91 years)), had higher HbA1c (5.6% (range 4.2-14%) versus 5.4% (range 3.9-14%), and more often diabetes (16% versus 5.5%), higher systolic BP 144 mmHg (range 72-286mmHg) versus 129 mmHg (70-222 mmHg), and higher diastolic BP 90 mmHg (range 42-150mmHg) versus 80 mmHg (range 36-140mmHg) than patients without albuminuria (p<0.001 for all). Patients with albuminuria had more often stage I and stage II hypertension (46% versus 24% and 28% versus 9% (p<0.001)), were more often HIV positive (11% versus 5.9% (p =0.032)), had more often a history of tuberculosis (11% versus 3.8% (p=0.001)), and had more acute infections (14% versus 7.4% (p=0.013)) than patients without albuminuria.”

  1. It is not clear what kind of samples were used for testing, random or first-morning specimen (page 2, line 87-89)? If it’s not standardized in this study, would it affect results? Another concern was the urine dipstick methods selection. Since 30mg/g is considered a clinically relevant cut-off, why did the authors used tests with two different detection limits? What were the selection criteria for this issue? This should be clearly stated in the method section. (Adv Chronic Kidney Dis 2011 Jul;18(4):243-8. doi: 10.1053/j.ackd.2011.03.002)

The authors thank for the comments. We used a random urine specimen for testing. In principle, the variability of albuminuria during the day can have an impact on the prevalence. However, the prevalence of albuminuria in our study was within the expected range for the region. [3-5] Therefore, the effect should be negligible. Further, since the analyses for the index and the reference tests were performed on the same sample, it is not expected that the random collection of samples will affect the concordance and agreement of the index and the reference tests.

In the method section on page 2 line 91 we added: “A random clean urine specimen was collected in all patients.”

In the limitation’s section of the discussion on page 13 line 463-466 we added: “Finally, due to the variability of albuminuria during the day, the random collection of urine samples could skew the prevalence rate,[6] but this should not influence the finding of the concordance and agreement between index and reference tests.”

We thank the reviewers for their comments regarding the different detection limits of the two index tests. We deliberately chose a highly sensitive immunochromatographic test and a less sensitive indicator dye-based test in order to investigate to what extent dipstick tests with different detection limits have an influence on possible screening algorithms.

We added in the method section on page 3 line 120-121: “Two index tests with different detection limits were chosen to analyse the influence of detection limits on models of albuminuria screening algorithms. “

  1. Because dipstick assays were carried out manually (page 3, line 93-96), basic method validation should be provided. This could be a comparison of negative and (1+) dipstick with ACR results. If applicable, standard addition methodology would be desirable (Journal of AOAC INTERNATIONAL, Volume 78, Issue 2, 1 March 1995, Pages 471–476, https://doi.org/10.1093/jaoac/78.2.471). The validation should prove that the manual readout was close to the manufacturer’s detection limit. Moreover, a single dipstick test could not be sufficient to detect proteinuria, and a double test would be needed to match with ACR result (Int J Environ Res Public Health. 2020 Jun; 17(12): 4195. doi: 10.3390/ijerph17124195).

We appreciate the comment and agree with the reviewer that this is an important point. Therefore, we illustrated in figure 3 the agreement/disagreement of the index Combur9-Test® and the Micral-Test® with the ACR reference test. Particularly, we compared negative and (1+) with ACR results for both dipstick tests. It is indeed the case that those who are negative or only weakly positive in either dipstick test, have mainly no albuminuria with an ACR < 30mg/g. This observation can serve as an indirect prove that the readout was close to the manufacturer`s detection limit.

In the results section on page 12 line 331 we added the figure caption title: “Fig. 4: Agreement of Combur9-Test® and Micral-Test® with the albumin-to-creatinine ratio (ACR)”

In the results section on page 12 line 329 we added the extended figure 4:

On page 12 line 332-334 we add the figure caption: “4A: Colorimetric urinary dipstick Combur9-Test® results compared to the albumin-to-creatinine ratio and to the  -Test®; 4B: Immunochromatographic Micral-Test® results compared to the albumin-to-creatinine ratio and to the colorimetric urinary dipstick Combur9-Test®;” ACR: albumin-to-creatinine ratio.”

On page 11 line 309-328 the results section was extended as follows:

“Summarized in figure 4A is the agreement of the semi-quantitative Combur9-Test® with the ACR reference test and the Micral-Test®. From the 627 (100%) patients with a negative Combur9-Test®, 96.6% (n=606) had a negative and 3.4% (n=21) a positive ACR and 22.8% (n=143) a negative and 77.2% (n=481) a positive Micral-Test®.  From the 240 (100%) patients with a 1+ (30mg/dL) Combur9-Test®, 84.6% (n=203) had a negative and 15.4% (n=37) a positive ACR and 5.5% (n=13) a negative and 94.5% (n=227) a positive Micral-Test®.  From the 70 (100%) patients with 2+ (100mg/dL) positive Combur9-Test®, 48.6% (n=34) had a negative and 51.4% (n=36) a positive ACR and 4.3% (n=3) a negative and 95.7% (n=67) a positive Micral-Test®. From the n=33 (100%) patients with a ≥3+ (≥300mg/dL) positive Combur9-Test®, 12.1% (n=4) had a negative and 87.9% (n=29) a positive ACR and 100% (n=33) a positive Micral-Test.

Summarized in figure 3B is the agreement of the semi-quantitative Micral-Test® with the ACR reference test and the Combur9-Test®. From the 158 (100%) patients with a negative Micral-Test®, 93.7% (n=148) had a negative and 6.3% (n=10) a positive ACR and 91% (n=143) a negative and 9% (n=15) a positive Combur9-Test®.  From the 448 (100%) patients with a 2mg/dL positive Micral-Test®, 98.2% (n=440) had a negative and 1.8% (n=8) a positive ACR and 87.3% (n=391) a negative and 12.7% (n=57) a positive Combur9-Test®. From the 209 (100%) patients with a 5mg/dL positive Micral-Test®, 89.5% (n=187) had a negative and 11.5% (n=22) a positive ACR and 31.6% (n=66) a negative and 68.4% (n=143) a positive Combur9-Test®. From the n=155 (100%) patients with a 10mg/dL positive Micral-Test®, 46.4% (n=72) had a negative and 53.6% (n=83) a positive ACR and 11% (n=17) a negative and 89% (n=138) a positive Combur9-Test®.”

Further, we agree that a single test is not sufficient to prove persistent proteinuria as pointed out in the section limitation page 15, line 457-459: “First, transient albuminuria (due to the day-to-day variability) could not be excluded due to the cross-sectional study design with a missing second urine sample for confirmation….”

Minor:

  1. Apart from NPV and PPV values, ROC analysis would enrich the results of your study.

We thank the reviewer for the valuable input. We have generated receiver operator curves (ROC) for both index tests against the ACR reference test with cut-off ≥30mg/g for albuminuria. Additionally, we have calculated area under the curve (AUC) measures. In the graph we included the AUC. We added to the new figure 2 to the results section on page 8 line 258

To the results section on page 7 line 247-248 we added: “This is further explored and illustrated in the receiver operator curves (ROC) and its corresponding area under the curve (AUC) values shown in figure 2.”

We add a figure caption on page 8 line 260-262: Figure 2: “Receiver operator curves (ROC) of index dipstick tests against albumin-to-creatinine ACR reference test with a cut-off ≥30mg/g for albuminuria. 2A: Combur9-Test® versus ARC reference test; 2B: Micral-Test® versus ARC reference test; AUC: Area under the curve.” 

  1. Please consider citing: Nephron 2018;140:169–174 DOI: 10.1159/000490954 ; Adv Chronic Kidney Dis 2011 Jul;18(4):243-8. doi: 10.1053/j.ackd.2011.03.002 ; Int J Environ Res Public Health. 2020 Jun; 17(12): 4195. doi: 10.3390/ijerph17124195

As suggested by the reviewer we cite “Viswanathan G, Upadhyay A: Assessment of Proteinuria. Advances in Chronic Kidney Disease 2011, 18(4):243-248.” (see page 15, line 465, ref 45)

References:

  1. Mancia G: Hypertension: Strengths and limitations of the JNC 8 hypertension guidelines. Nat Rev Cardiol 2014, 11(4):189-190.
  2. WHO: Haemoglobin concentrations for the diagnosis of anaemia and assessment of severity. Vitamin and Mineral Nutrition Information System. Geneva, World Health Organization. (WHO/NMH/NHD/MNM/11.1). http://www.who.int/vmnis/indicators/haemoglobin.pdf (Accesses Feb. 2019).
  3. Hill NR, Fatoba ST, Oke JL, Hirst JA, O'Callaghan CA, Lasserson DS, Hobbs FD: Global Prevalence of Chronic Kidney Disease - A Systematic Review and Meta-Analysis. PLoS One 2016, 11(7):e0158765.
  4. Peck R, Baisley K, Kavishe B, Were J, Mghamba J, Smeeth L, Grosskurth H, Kapiga S: Decreased renal function and associated factors in cities, towns and rural areas of Tanzania: a community-based population survey. Trop Med Int Health 2016, 21(3):393-404.
  5. Stanifer JW, Maro V, Egger J, Karia F, Thielman N, Turner EL, Shimbi D, Kilaweh H, Matemu O, Patel UD: The Epidemiology of Chronic Kidney Disease in Northern Tanzania: A Population-Based Survey. PLOS ONE 2015, 10(4):e0124506.
  6. Viswanathan G, Upadhyay A: Assessment of Proteinuria. Advances in Chronic Kidney Disease 2011,18(4):243-248.

Reviewer 2 Report

The authors designed a cross-sectional study in little over 1000 subjects to test the validity of a semi-quantitative standard colorimetric 60 indicator-dye based multi dipstick (Combur9-Test®) and an albumin specific 61 immunochromatographic assay (Micral-Test®) compared to the quantitative ACR reference 62 diagnostic test.

The study is well designed and performed and the results are clearly presented according to the STARD Guidelines for diagnostic accuracy studies. 

The only suggestion would be to add the issue of the limitation of the diagnostic tests for albuminuria due to its high day-to-day variability and the effect of age and gender 

Author Response

Review II

The authors designed a cross-sectional study in little over 1000 subjects to test the validity of a semi-quantitative standard colorimetric 60 indicator-dye based multi dipstick (Combur9-Test®) and an albumin specific 61 immunochromatographic assay (Micral-Test®) compared to the quantitative ACR reference 62 diagnostic test.

The study is well designed and performed and the results are clearly presented according to the STARD Guidelines for diagnostic accuracy studies. 

The only suggestion would be to add the issue of the limitation of the diagnostic tests for albuminuria due to its high day-to-day variability and the effect of age and gender.

The authors thank for the comments. We added the following sentence on section limitation, page 15, line 457-459: “First, transient albuminuria (due to the day-to-day variability) could not be excluded due to the cross-sectional study design with a missing second urine sample for confirmation….”